# NIR and Reduction Dual-Sensitive Polymeric Prodrug Nanoparticles for Bioimaging and Combined Chemo-Phototherapy

**DOI:** 10.3390/polym14020287

**Published:** 2022-01-11

**Authors:** Shuying Li, Yanjuan Wu, Xiukun Xue, Siyuan Liu

**Affiliations:** Shandong Provincial Key Laboratory of Molecular Engineering, School of Chemistry and Chemical Engineering, Qilu University of Technology (Shandong Academy of Sciences), Jinan 250353, China; L15662692779@163.com (S.L.); Jasonxue@yeah.net (X.X.); liusy@qlu.edu.cn (S.L.)

**Keywords:** combined chemo-phototherapy, Pt(IV) prodrug, nanoparticle, bioimaging

## Abstract

The combination of chemotherapy, photothermal therapy (PTT) and photodynamic therapy (PDT) based on a single nanosystem is highly desirable for cancer treatment. In this study, we developed a versatile Pt(IV) prodrug-based nanodrug, PVPt@Cy NPs, to realize synchronous chemotherapy, PDT and PTT and integrate cancer treatment with bioimaging. To construct PVPt@Cy NPs, the amphiphilic Pt(IV)-based polymeric prodrug PVPt was synthesized by a facile one-pot coupling reaction, and then it was used to encapsulate an optotheranostic agent (HOCyOH, Cy) via hydrophobic interaction-induced self-assembly. These NPs would disaggregate under acidic, reductive conditions and NIR irradiation, which are accompanied by photothermal conversion and reactive oxygen species (ROS) generation. Moreover, the PVPt@Cy NPs exhibited an enhanced in vitro anticancer efficiency with 808-nm light irradiation. Furthermore, the PVPt@Cy NPs showed strong NIR fluorescence and photothermal imaging in H22 tumor-bearing mice, allowing the detection of the tumor site and monitoring of the drug biodistribution. Therefore, PVPt@Cy NPs displayed an enormous potential in combined chemo-phototherapy.

## 1. Introduction

Cancer is still one of the major health threats in the world. Chemotherapy, one of conventional treatment modalities, is a remarkable success in clinical practice [1]. Despite the significant achievements in chemotherapy, it is still facing apparent and nonnegligible challenges, such as non-specific distribution, rapid metabolic clearance of drugs as well as the possible development of multidrug resistance (MDR) [2,3]. Nowadays, new therapeutic methods such as photodynamic therapy (PDT), photothermal therapy (PTT), immunotherapy, gene therapy and chemodynamic therapy (CDT) have drawn great attention, with success to a certain extent [4,5]. However, the therapeutic outcomes of monotherapy alone are generally unsatisfactory because of the complexity, diversity, metastasis, recurrence and resistance of cancers [2,6,7]. Combinatorial multimodal therapy is considered a promising and prospective approach to improve antitumor efficiency and minimize systematic cytotoxicity.

Among various multimodal therapies, chemo-phototherapy has been extensively investigated [8,9,10,11,12]. Phototherapy, mainly including PDT and PTT, is expected to possess many exclusive advantages, such as a noninvasive nature, high spatiotemporal controllability, diminutive side effects and negligible drug resistance [5,13,14]. On the other hand, PDT and PTT also face some bottleneck problems, including tumor hypoxia, insufficient light penetration depth and the heat resistance of the tumor cells [15]. Thus, combined chemo-phototherapy is a satisfactory approach to circumvent the above problems and achieve effective treatment of cancers. Consequently, various types of nanocarriers, including mesoporous silica nanoparticles [16,17], metal nanoparticles [18,19,20], molybdenum disulfide (MoS_2_) nanomaterials [21,22,23], polydopamine nanoparticles [24,25,26,27] and self-assembled polymeric nanoparticles [28,29], were used to integrate chemotherapeutic drugs and photosensitizer (PS) or photothermal agents into one multifunctional system for chemo-phototherapy. Notably, polymeric nanoparticles as delivery systems are very attractive, especially owing to their biodegradability, biocompatibility, versatility and capability of sustained release, which are crucial features for improving the pharmokinetics of guest cargos and minimizing the negative side effects [30].

Near-infrared (NIR) light (700–900 nm) with deep tissue penetration, low light scattering, high light energy utilization and low phototoxicity is the preferred and desired light source for clinical applications. Cyanine dyes, like cypate or indocyanine green (ICG), are typical NIR dyes and were generally co-loaded with various chemotherapeutic drugs to fabricate this kind of combined therapy system due to the intrinsic multiple functionalities, such as low cytotoxicity, high photothermal conversion efficiency, reactive oxygen species (ROS) generation, fluorescence imaging, photothermal imaging and photoacoustic imaging [31,32,33]. Wang et al. synthesized a polymer nanoparticle named (IR780/DOX)@PTK, which was activated and degraded under an NIR light, and IR780 and DOX were co-loaded into an ROS-sensitive PTK polymer by physical encapsulation. (IR780/DOX)@PTK has remarkable photodynamic and photothermal effects and minimal drug toxicity [34]. Wang et al. reported on ICG and paclitaxel (PTX) co-loaded, bioinspired particulate, which was formulated with native high-density lipoproteins (pHDL) and decorated with iRGD targeting moiety [35]. After accumulation in tumor cells, the ROS and hyperthermia were generated by ICG upon NIR irradiation. Then, the pHDL would rapidly collapse, resulting in compact structure disassembly and intracellular PTX burst drug release. This system could facially realize fluorescence imaging-guided precision chemo-phototherapy. Overall, previous studies confirmed that cyanine dyes and chemotherapeutic drugs co-loaded with nanoparticles showed application potency for chemo-phototherapy. Nonetheless, both cyanine dyes and chemotherapeutic drugs were usually incorporated in nanocarriers via electrostatic or hydrophobic interaction, resulting in a limited drug encapsulation capacity and unsatisfactory burst release of chemotherapeutic drugs. Therefore, it is of great significance to further develop effective polymeric nanodrugs for more effective combined chemo-phototherapy.

In this work, we develop a novel, multifunctional drug delivery system PVPt@Cy NPs based on the co-assembly of Pt(IV)-based polymeric prodrug (PVPt) and a modified cyanine dye 1-(2-hydroxyethyl)-2-((E)-2-((E)-3-((E)-2-(1-(2-hydroxyethyl)-3,3-dimethylindolin-2-ylidene)ethylidene)-2-chlorocyclohex-1-en-1-ly)vinyl)-3,3-dimethyl-3H-indol-1-ium bromide (HOCyOH, denoted as Cy) for realizing combined chemotherapy, PTT and PDT cancer treatment (Figure 1). The PVPt was prepared by a facile one-pot coupling reaction, where α-tochoferol was used to modify the hydrophobic ability, PEG was hydrophilic segment, and Pt(IV) exerted an anticancer effect. DLS measurement indicated that the self-assembled PVPt@Cy NPs were stable under a physiological pH. However, the NPs would disaggregate in acidic reduction conditions and 808-nm irradiation. With 808-nm laser irradiation, PVPt@Cy NPs could not only generate local hyperthermia for PTT but also produce singlet oxygen (^1^O_2_) for PDT. Furthermore, PVPt@Cy NPs showed reduction-triggered Pt release due to the inherent reduction-sensitive property of Pt(IV). The cytotoxicity of PVPt@Cy NPs against HeLa cells was determined by a standard methyl thiazolyl tetrazolium (MTT) assay. In addition, in vivo fluorescence imaging and photothermal imaging was studied. The PVPt@Cy NPs have great potential for imaging-guided combination therapy.

## 2. Materials

### 2.1. Materials

Cisplatin (99.8%) was purchased from Shandong Boyuan Pharmaceutical Co., Ltd. (Jinan, China). Poly(isobutylene-alt-maleic anhydride) (PMA) (6000 Mw, 85%) and Hoechst 33258 (98%) were obtained from Sigma-Aldrich (Shanghai, China). (R)-2,5,7,8-Tetramethyl-2-((4R,8R)-4,8,12-trimethyltridecyl)chroman-6-ol (α-tochoferol, 96%) was purchased from Bide Pharmaceutical Technology Co., Ltd. (Shanghai, China). Succinic anhydride (SA, 99%), glutathione GSH (reduction type, 98%), cyclohexanone (99%, AR) and methoxypolyethylene glycol (mPEG_2000_-OH) were purchased from Aladdin Chemical Co., Ltd. (Shanghai, China). Phosphate-buffered saline (PBS), high-glucose medium (DMEM), Pen-Strep solution (Penicillin: 100 U/mL, Streptomycin: 0.1 mg/mL), fetal bovine serum (FBS) and trypsin EDTA solution A were purchased from Biological Industries (Shanghai, China). Sodium phosphate monobasic dihydrate (NaH_2_PO_4_·2H_2_O, AR) and 1,3-diphenylisobenzofrofuran (DBPF, 97%) were purchased from Maclin Reagent (Shanghai, China). POCl_3_ (95%), 2-bromoethanol (98%, AR) and 2,3,3-trimethylindolenine (98%, AR) were purchased from Anergy Chemical Reagent (Shanghai, China). Hydrogen peroxide solution (H_2_O_2_, 30 wt% aqueous solution), disodium hydrogen phosphate dodecahydrate (Na_2_HPO_4_·12H_2_O, AR), N,N-dimethylformamide (DMF, AR), dichloromethane (DCM, AR) and dimethyl sulfoxide (DMSO, AR) were purchased from Sinopharm Group Chemical Reagents Co., Ltd. (Shanghai, China). Anhydrous reagents were dried with calcium hydroxide (CaH_2_) for 1 day, and anhydrous solvent was obtained by decompression distillation. All chemical reagents were used directly unless otherwise indicated.

Moreover, according to the previous literature, we synthesized 1-(2-hydroxyethyl)-2-((E)-2-((E)-3-((E)-2-(1-(2-hydroxyethyl)-3,3-dimethylindolin-2-ylidene)ethylidene)-2-chlorocyclohex-1-en-1-ly)vinyl)-3,3-dimethyl-3H-indol-1-ium bromide (Cy) [36]. Additionally, cisplatin was oxidized to DHP [37,38], and then we synthesized HO-Pt-COOH with DHP and SA [39]. (Please refer to Supporting Information for the specific synthesis route.)

### 2.2. Measurement

The ^1^H-NMR spectrum was recorded with an AVANCE II 400 MHz NMR spectrometer (Bruker, Fällanden, Switzerland) using DMSO-d_6_ or D_2_O as the solvent at room temperature. The Fourier Transform Infrared (FT-IR) spectra were recorded using a Nicolet IS10 spectrometer(Thermo Fisher Scientific, Waltham, MA, USA). Fluorescence spectra were obtained on an F97 Pro (Lengguang Tech. Shanghai, China) fluorescence spectrophotometer. The UV–visible light absorption spectra were measured on a UV2800S UV–visible spectrophotometer(Shanghai Yongneng Investment Management Co., Ltd., Shanghai, China). The diameter and potential of the nanoparticles were measured using a Zetasizer laser particle size analyzer (Malvern Zetasizer Nano-ZS 90, Malvern Panalytical company, Malvern, UK). The size and morphology of the nanoparticles were recorded using a Transmission Electron Microscope (TEM) on the controller of a JEM 2100(JEOL Ltd., Tokyo, Japan) electron microscope. Confocal laser scanning microscopy (CLSM) micrographs were recorded using a Zeiss LSM880+ Fast Airyscan(Carl Zeiss AG, Oberkochen, Germany). The 808-nm infrared laser (Xi’an Reaser Electronic Technology Co., Ltd., Xi’an, China) measured the photothermal and photodynamic performance of the prepared nanodrugs. Drug release assays were performed when incubated with an SHA-B oscillator. The toxicity of different samples to cells was determined on a Synergy Neo2 microplate reader(BioTek Instruments, Inc., Winooski, VT, USA) using MTT. The measured cell fluorescence intensity could be recorded by FACS Calibur flow cytometry (Becton, Dickinson and Company, Franklin Lakes, NJ, USA). The platinum content was measured by the inductively coupled plasma mass spectrometer (ICP-MS, Thermoscientific, Waltham, MA, USA) and inductively coupled plasma optical emission spectrometer (ICP-OES, Thermoscientific, Waltham, MA, USA).

## 3. Methods

### 3.1. Synthesis and Formulation of the PVPt@Cy NPs

#### 3.1.1. Synthesis of the Pt(IV)-Based Polymer Prodrug (PVPt)

PMA (0.21 g, 0.035 mmol) and c,c,t-[Pt(NH_3_)_2_Cl_2_(OH)(O_2_CCH_2_CH_2_CH_2_CO_2_H)] (HO-Pt-COOH, 0.33 g, 0.7 mmol) were dissolved in dry DMF (10 mL). The reaction solution was kept at 60 °C under the protection of N_2_ gas for 48 h. Then, α-tochoferol (0.135 g, 0.3 mmol) and mPEG_2000_-OH (0.349 g, 0.17 mmol) were added into the above solution, and the mixture was allowed to stir for another 48 h in the dark, followed by precipitation in 200 mL of diethyl ether. The obtained crude product was dialyzed against Milli-Q water (MWCO: 3500 Da) for 48 h for further purification, and then the suspension was freeze-dried to give PVPt, yielding 0.75 g (73%).

#### 3.1.2. Preparation of Cy-Loaded PVPt Nanoparticles (Denoted as PVPt@Cy NPs)

PVPt@Cy NPs were prepared by a nanoprecipitation method [34]. In general, PVPt (10 mg) and Cy (1 mg) were dissolved in 10 mL of DMSO and stirred for 12 h at room temperature in the dark. Then, the solution was gradually added in 9 mL of deionized water under stirring. The mixture was stirred for 4 h at room temperature. Subsequently, the suspension was transferred into a dialysis bag and dialyzed against deionized water for 48 h. Finally, the dialysate was lyophilized to obtain PVPt@Cy NPs. The PVPt NPs were prepared similarly.

#### 3.1.3. Drug Loading Capacity

In the subsequent analysis, the freeze-dried PVPt@Cy NPs (2 mg) were dissolved in 40 mL of DMSO, and the absorbance of Cy was measured by a UV−vis spectrometer [31]. The Cy loading capacity (DLC) and Cy loading efficiency (DLE) could be calculated according to the following formula:DLC (%) = [Weight of Cy in PVPt@Cy NPs/total mass of PVPt@Cy NPs] × 100%,(1)
DLE (%) = [Weight of Cy in PVPt@Cy NPs/initial feed amount of Cy] × 100%,(2)

#### 3.1.4. Stability and Responsiveness of PVPt NPs and PVPt@Cy NPs

The stability and responsiveness of the prepared PVPt NPs were determined by DLS measurement, and the variation in the particle size and size distribution were monitored over time. In detail, PVPt NPs were incubated in phosphate-buffered saline (PBS, pH = 7.4), PBS 7.4 with 10 mM GSH, acetate buffer solution (ABS, pH = 5.0) or ABS 5.0 with 10 mM GSH. At a predetermined time point, the samples were characterized by DLS measurement. Moreover, the responsiveness of PVPt@Cy NPs to NIR irradiation was also characterized by DLS. The suspension of PVPt@Cy NPs was irradiated by an 808-nm laser for 30, 60 or 300 s, and the changes in the particles and size distribution were determined.

#### 3.1.5. In Vitro Photothermal and Photodynamic Performance of PVPt@Cy NPs

PVPt@Cy NPs at Cy concentrations of 0, 20, 40 and 50 mg/L were treated with 808-nm light irradiation at the power density of 1.0 W/cm^2^ for 5 min. The increased temperatures of the samples were recorded in detail separately every 10 s in the EP tubes. Additionally, the PVPt@Cy NPs solutions (containing 20 mg/L Cy) were subjected to 808-nm laser irradiation at various power densities (0.5, 1.5 and 2.0 W/cm^2^), and the corresponding temperature variations were measured every 10 s.

To investigate the ^1^O_2_ generation capability, DPBF as a ^1^O_2_ probe was added into the PVPt@Cy NPs solution (0.25 mg/L), and the mixture was irradiated by an 808-nm laser with a fixed power density (0.5 W/cm^2^). At predetermined time intervals, the samples were measured on an ultraviolet–visible spectrophotometer to detect the absorbance of DPBF at 425 nm, which could reflect the production of ^1^O_2_. Similarly, samples without irradiation were taken at the same time interval for characterization.

#### 3.1.6. In Vitro Drug Release

The in vitro Pt release from PVPt@Cy NPs was investigated by a dialysis method. In general, the lyophilized PVPt NPs (5 mg) were redispersed in 5 mL of PBS (pH = 7.4), PBS (pH = 7.4) with 10 mM GSH, ABS (pH = 5.0) or ABS (pH = 5.0) with 10 mM GSH, respectively. Then, the prepared solutions were transferred into a dialysis bag (MWCO: 3500 Da) and immersed in 45 mL of the corresponding buffer solution. The system was placed in an oscillator at 37 °C. At desired time points, 2 mL of dialysate was sampled to determine the content of Pt by ICP-MS, and 2 mL of the corresponding fresh buffer solution was replenished.

### 3.2. Biological Activity

#### 3.2.1. Cell Culture

The human cervical cancer cells (HeLa) used in the experiment were bought from the Institute of Biochemistry and Cell Biology at the Chinese Academy of Sciences (Shanghai, China). We prepared the DMEM with a volume fraction of 10% FBS and 1% Pen-Strep solution (100 IU/ mL). After inoculating the HeLa cells in the prepared DMEM, we incubated them in a 37 °C incubator with 5% CO_2_.

#### 3.2.2. Intracellular ROS Assay

Under 808-nm light irradiation, we used DCFH-DA as a probe to detect the intracellular generation of ROS from PVPt@Cy NPs. In detail, the HeLa cells were inoculated in a sterile 6-well plate with coverslips and cultured for more than 24 h. After the cell density reached the ideal state, PVPt@Cy NPs and the free Cy with the equivalent concentration (5 mg/L) were added and incubated for 4 h. Then, the cells were incubated with DCFH-DA (10 mg/L) for another 20 min and treated by an 808-nm laser (0.5 W/cm^2^) for 2 min. The cells without 808-nm irradiation were used as a control. Then, the fluorescence imaging could be obtained by an inverted fluorescence microscope. At the same time, the HeLa cells were treated under the same conditions, and the fluorescence intensity was quantitatively analyzed using a flow cytometer.

#### 3.2.3. In Vitro Intracellular Uptake and Intracellular Distribution

The in vitro cellular uptake efficiency and intracellular drug distribution of HeLa cells to PVPt@Cy NPs were recorded by a Zeiss LSM880+ Fast Airyscan laser confocal electron microscope. A total of 2 × 10^5^ HeLa cells were seeded in 6-well plates with coverslips and cultured in an incubator for more than 24 h. Then, the cells were treated with PVPt@Cy NPs (10 mg/L of Cy) and incubated for another 2 and 12 h. After incubation, the cells were rinsed gently with PBS 3 times and then fixed with a 4% (*w*/*v*) paraformaldehyde solution in PBS for 20 min. After being washed with PBS, the cell nuclei were stained with Hoechst 33258 (10 mg/L) for 10 min. Finally, the cells were additionally rinsed with PBS, and the intracellular uptake of the PVPt@Cy NPs was observed using CLSM. Simultaneously, flow cytometry analysis was performed for quantitative evaluation. The HeLa cells were cultured in a 6-well plate at a concentration of 2 × 10^5^ cells/well. After adhering to the wall, the cells were treated with 2 mL of DMEM containing PVPt@Cy NPs, and the final concentration of Cy was 10 mg/L. Thereafter, the cells were washed with cold PBS and trypsinized. The HeLa cells were harvested by centrifugation and resuspended in PBS for analysis.

#### 3.2.4. In Vitro Cytotoxicity

We used the MTT method to assess the in vitro cytotoxicity of the free Cy, cisplatin, DHP, DHP + Cy and PVPt@Cy NPs. Generally, the HeLa cells in the logarithmic growth phase were digested with trypsin and seeded into a 96-well plate at a concentration of 4 × 10^5^ cells/well. After incubation overnight, the medium was replaced with fresh DMEM medium, which contained various drugs at certain Pt concentrations (0.31, 0.63, 1.25, 2.5, 5 and 10) or Cy concentrations (0.15, 0.3, 0.6, 1.2, 2.4 and 4.8) (mg/L), separately. After 6 h of incubation, for the 808-nm light irradiation groups, the HeLa cells were irradiated by 808-nm light (1.0 W/cm^2^) for 5 min before culturing in the dark for another 48 h. Subsequently, 20 μL of MTT solution (5 g/L) was added to each well, and the incubation was kept for 4 h. Then, the medium was replaced by 150 μL of DMSO to dissolve the formazan products. Finally, the absorbance of the formazan crystal was determined at 490 nm using a microplate reader. The cell viability could be obtained by the following formula:cell viability (%) = [OD of treated cells /OD of control cells] × 100% (optical density: OD),(3)

#### 3.2.5. Animal Models

All of the in vivo study was performed according to the guidelines for the care and use of animals published by the regional animal committee. The 5−6-week-old male Kunming (KM) mice were bought from Shanghai SLAC Laboratory Animal Co., Ltd (Shanghai, China). The H22 subcutaneous tumor model was developed by subcutaneously injecting H22 cells (5 × 10^6^, 0.1 mL PBS) into the left legs of the mice. The H22 tumor-bearing mice were applied to the following study when the volume was larger than 200 mm^3^.

#### 3.2.6. In Vivo Bioimaging

We used near-infrared fluorescence imaging to evaluate the biodistribution of the PVPt@Cy NPs in mice. H22 tumor-bearing mice were intravenously injected with 100 μL of free Cy (5 mg/kg) or an equivalent concentration of PVPt@Cy NPs. A near-infrared fluorescence imaging assay was performed at 1, 5, 10 and 24 h after injection. Then, the mice were euthanized, and the tumors and major organs (i.e., heart, liver, spleen, lungs and kidney) were harvested for near-infrared fluorescence imaging.

In addition, photothermal imaging of the PVPt@Cy NPs was conducted. Specifically, H22 tumor-bearing mice were injected with 100 μL of PBS, Cy (5 mg/kg) and PVPt@cy NPs via the tail vein. After 24 h, an 808-nm NIR (2 W/cm²) laser was used to irradiate the tumor site for 4 min, and the photothermal images were taken by a FLIR pro thermal camera.

#### 3.2.7. Statistical Analysis

All data were the representative results on the basis of at least three independent experiments and represented as the mean ± standard deviation (SD). Significant differences between various groups were calculated by a Student’s *t*-test (where ns stands for no significance and * *p* < 0.05, ** *p* < 0.01, *** *p* < 0.001 and **** *p* < 0.0001 were considered statistically significant).

## 4. Results and Discussion

### 4.1. Preparation of Polymer Prodrug PVPt and Cy-Loaded PVPt Nanoparticles (PVPt@Cy NPs)

At first, the c,c,t-[Pt(NH_3_)_2_Cl_2_(OH)(O_2_CCH_2_CH_2_CH_2_CO_2_H)] (HO-Pt-COOH, Pt (IV) prodrug) was prepared according to our previous work [37]. The synthesis route is displayed in the Supporting information, and the ^1^H NMR spectrum is shown in Appendix A. The cyanine dye containing two hydroxyl groups (HOCyOH and Cy) was synthesized according to the literature with a little modification [36,40]. In the ^1^H NMR spectrum (Appendix A), when the two peak signals at δ = 8.3 and 6.5 ppm were ascribed to the protons from the bridge between cyclopentane and cyclohexene, a peak at 1.6 ppm belonging to the methyl group appeared, and characteristic peaks of the benzene ring at 7.1–7.4 ppm were also observed. These results demonstrate that the Cy was successfully prepared. Then, for the preparation of polymer prodrug PVPt, HO-Pt-COOH, α-tochoferol and mPEG_2K_-OH were used to modify the PMA via a one-pot esterification reaction. The PVPt was synthesized, and the synthesis route is shown in Figure 2. Furthermore, the successful conjugation of HO-Pt-COOH, α-tochoferol and mPEG_2K_-OH was confirmed by the ^1^H NMR spectrum (Figure 1A) and FT-IR spectrum of PVPt (Appendix A). The ^1^H NMR spectrum of PVPt showed the signal overlap of VE, the characteristic proton signal of PEG which could be observed at δ = 3.5 ppm and the chemical shifts at δ = 2.5 and 6.5 ppm belonging to the HO-Pt-COOH. Compared with the ^1^H NMR spectrum in DMSO d_6_, only the peak of the PEG (–OCH_2_CH_2_, d = 3.49 ppm) appeared in D_2_O, which indicates that PVPt could self-assemble into nanoparticles in an aqueous medium (Figure 1B). Moreover, the molecular weight of PVPt was measured by GPC. As demonstrated in Figure 1C, the GPC result of the PVPt demonstrated that the average molecular weight and PDI of PVPt were about 15.2 kg/mol and 1.13, respectively. The Pt loading capacity and efficiency in PVPt were measured by ICP-OES to be 18.1% and 94.2%, respectively.

The amphiphilic nature of PVPt enables the formation of nanoparticles in an aqueous medium. The PVPt and PVPt@Cy NPs were facially prepared by a well-established nanoprecipitation method. Then, the DLS and TEM techniques were used to determine the size, size distribution and morphology of the prepared nanoparticles. As shown in Figure 2A, the average hydrodynamic sizes and particle size distribution (PDI) of PVPt NPs were determined to be 59 nm and 0.181, respectively. After the encapsulation of Cy, both the size and PDI increased (Figure 2B). After lyophilization, the size of the redispersed PVPt@Cy NPs was slightly increased (Appendix A). However, the hydrodynamic size of the NPs was always kept below 200 nm. The little nanoparticle sizes were favorable for effective tumor accumulation and then improving the anticancer efficiency [41]. Interestingly, PVPt could self-assemble into nanomicelles in the TEM image (Figure 2C), and nanomicelles were also observed for the PVPt@Cy NPs (Figure 2D). The diameters measured by DLS were slightly higher than those results by TEM. This phenomenon is common, which can be ascribed to the hydration effect of the PEG shell in an aqueous medium [42]. In addition, the zeta potentials of PVPt and PVPt@Cy NPs were −32.2 and −37.9 mV, respectively (Appendix A), which favored the minimization of non-specific protein absorption during blood circulation [43]. In addition, using the calibration curve prepared by a standard solution with a known Cy concentration, the quality of the Cy in the NPs was quantified with the UV–vis spectrum. The drug loading content of Cy for the PVPt@Cy NPs was 8.7%.

### 4.2. Stability Evaluation

It is well known that the physiological stability of the nanodrug is one of the significant prerequisites for anticancer application [15,41]. The stability of the PVPt and PVPt@Cy NPs in different conditions was examined by monitoring the size and size distributions of these particles. The results illustrated in Figure 3A show that the PVPt NPs still maintained their sizes and size distributions for up to 48 h in PB (pH = 7.4), indicating the high stability of the prepared nanodrugs. Importantly, high stability of the nanodrugs would be favorable for in vivo blood circulation after intravenous injection. To mimic the acidic and reductive microenvironment in the tumor cells, the dispersion of PVPt NPs was incubated in PBS (PH = 7.4), ABS (PH = 5.0), PBS (PH = 7.4) with 10 mM GSH and ABS (pH = 5.0) with 10 mM GSH. As can be seen in Figure 3B, the size and size distribution of the PVPt NPs underwent rapid and remarkable changes at a pH of 5.0, in which the average size increased from ca. 59 to 225 nm in 24 h. The changes were likely attributed to the abundant carboxyl groups of the nanodrug. Certainly, similar phenomena were observed for the nanodrugs treated at a pH of 7.4 with 10 mM GSH and a pH of 5.0 with 10 mM GSH (Figure 3C,D), which should have been due to the Pt release and reassembly in these media. These results indicate that the acidic pH and tumor-overexpressed GSH could induce PVPt NP disassembly.

Cy is a NIR dye which can act as a medicament for PDT and PTT. Therefore, the photosensitive behavior of the PVPt@Cy NPs was further investigated via DLS measurement. After exposure to an 808-nm light for 30 s, the average size of the PVPt@Cy NPs increased from 98 nm to 144.4 nm. In particular, the size and size distribution varied over the irradiation time (Figure 3E). The above results demonstrate that the PVPt@Cy NPs were very stable within 48 h, which was conducive to blood circulation and drug accumulation. However, the stability was damaged under the characteristic conditions of the tumor microenvironment. Next, the PVPt@Cy NPs released chemotherapeutic drugs and Cy. We believe that the PVPt@Cy NPs system can precisely control the release of drugs at the tumor site, thereby reducing the damage of chemotherapeutic drugs to normal tissues. The combined treatment of chemotherapy and PTT and PDT could be realized under near-infrared irradiation.

### 4.3. Photothermal and Photodynamic Properties

Intially, the photophysical property of the UV–vis absorption spectrum for the PVPt@Cy NPs was further characterized. As shown in Figure 4A, both the free Cy and PVPt@Cy NPs had a strong absorption band near 800 nm, and the absorption of PVPt@Cy NPs was slightly red-shifted, which revealed the encapsulation of Cy in the NPs [2]. The strong and broad absorbance of Cy from 700 to 900 nm encouraged us to study the photothermal ability of the PVPt@Cy NPs. The PVPt@Cy NPs at different Cy concentrations were exposed to an 808-nm NIR laser (1 W/cm^2^) for 5 min. Figure 4B obviously demonstrates the gradual concentration and irradiation time-dependent photothermal performance of the PVPt@Cy NPs. After irradiating for 5 min, the temperature of the PB solution negligibly changed, while the PVPt@Cy NPs at different Cy concentrations (0, 20, 40 and 50 mg/L) increased from 20.1 to 21.4, 31.7, 39.7 and 43.8 °C, respectively. Furthermore, the temperature variation (ΔT) of the PVPt@Cy NPs at a concentration of 20 mg/L Cy, after irradiation at four different power densities (0.5, 1, 1.5 and 2 W/cm^2^) for 5 min, could be found to be about 5.7, 11.8, 24.63 and 28.4 °C, respectively. Sanchez-Ramirez et al. synthesized poly(lactic-co-glycolic) nanoparticles (PLGA NPs), which are used to encapsulate the chemo-drug carboplatin (CP) and the NIR photosensitizer ICG [44]. When the system was exposed to 808-nm (2 W/cm^2^) laser irradiation for more than 10 min, the temperature variation (ΔT) reached about 24 °C. It can be seen that ΔT in our work could reach 28.4 °C under the same power (Figure 4C). The colors of the thermal images (Figure 4D) measured via the IR thermal camera were nearly consistent with the results of the photothermal curves (Figure 4C).

Additionally, Cy can generate ROS in the presence of 808-nm NIR light irradiation [22]. To confirm the PDT potentials of the PVPt@Cy NPs, the ROS generation was determined using DPBF as the ^1^O_2_ specific probe, whose absorption peaks at 425 nm decreased irreversibly with the generation of ^1^O_2_. As shown in Figure 4E and Appendix A, there was no significant difference in DPBF absorption in the NP solution. However, when the system was irradiated with an 808-nm laser, the absorbance of the DPBF gradually decreased, indicating the quick and sufficient generation of ^1^O_2_ (Figure 4E). Overall, the studies demonstrated that the PVPt@Cy NPs exhibited an outstanding photothermal conversion and ROS generation ability. PVPt@Cy NPs could serve as a remarkable photothermal agent for PTT and an effective photosensitizer agent for PDT.

The above results showed that the PVPt@Cy NP solution could show a good concentration dependence under 808-nm NIR laser radiation and showed a good photothermal effect. An elevated temperature in a relatively short period of time can cause substantial cell damage, which is of great significance for the combined treatment of cancer. At the same time, the production of ^1^O_2_ can also promote cell apoptosis at the tumor site and improve the efficiency of the combined treatment.

### 4.4. In Vitro Pt Release

Benefiting from the GSH-induced cleavage of Pt (IV), PVPt@Cy NPs were fabricated for GSH-triggered drug release. In the experiment, the release medium was the buffer of a different pH (pH 7.4 and 5.0) with or without 10 mM GSH, which simulated the pH of normal tissues and the microenvironment of tumor cells. As shown in Figure 5A, only 12% of the Pt was released at a pH of 7.4 in around 48 h, indicating that the prepared nanoparticles were stable in the neutral condition, which was beneficial to blood circulation. Moreover, the cumulative drug release and release rate of Pt in the PVPt@Cy NPs at a pH of 5.0 were similar to that at a pH of 7.4. Furthermore, at a pH of 7.4 with 10 mM GSH, the release of Pt was dramatically accelerated to 77%, owing to the reduction in Pt (IV) and the swelling of the NPs. Additionally, the Pt release rate at a pH of 5.0 with 10 mM GSH was slightly faster, and 83% platinum was released from the PVPt@Cy NPs. These results suggest that PVPt@Cy NPs could specifically respond to the reductive microenvironment to mediate controllable Pt release.

### 4.5. Intracellular ROS Detection and Cellular Uptake

In order to explore whether PVPt@Cy NPs could be effectively absorbed by HeLa cells, CLSM was applied to qualitatively investigate the cellular uptake and intracellular localization. As illustrated in Figure 5B, after incubation with free PVPt@Cy NPs for 2 h, the weak red fluorescence of Cy was found in the cytoplasm of the HeLa cells. This might be due to the slow endocytosis pathway of NPs [45]. As expected, stronger Cy fluorescence intensities were observed with the incubation time prolonged to 12 h. Notably, a red fluorescence signal was found in both the cytoplasm and nucleus, indicating that the Cy was partially released and translocated into the nuclei. The quantified fluorescence intensity by flow cytometer analysis could further support the CLSM observation (Appendix A). These results indicate that PVPt@Cy NPs not only delivered the drugs efficiently but also had excellent NIR imaging capabilities.

To evaluate the intracellular ROS generation property, DCFH-DA was used as a probe, which could be oxidized into the green fluorescent dye 2^!^,7^!^-dichlorofluorescein (DCF) by the generated ROS. As shown in Figure 5C, the free Cy, PVPt@Cy NPs and control groups exhibited quite weak fluorescence signals in the HeLa cells without 808-nm irradiation, mainly due to the low ROS level. In contrast, much stronger green fluorescence could clearly be seen in the free Cy and PVPt@Cy NPs groups treated with 808-nm laser irradiation, suggesting that the intracellular ROS levels were elevated. For quantitative analysis of the intracellular ROS generation, flow cytometry analysis was utilized with the HeLa cells. The results of the flow cytometry also displayed intracellular trends of the fluorescence intensity similar to those obtained by inverted fluorescence microscopy (Appendix A). These investigations further demonstrated that PVPt@Cy NPs could trigger high intracellular ROS levels for possible PDT with NIR light irradiation.

### 4.6. In Vitro Study of Intracellular Cytotoxicity

The potential cytotoxicity of the Cy-loaded polymeric prodrug PVPt@Cy NPs was assessed against the HeLa cells by a standard MTT assay in vitro. A series of cisplatin, Cy, DHP, DHP + Cy and PVPt@Cy NPs with equivalent drug concentrations were co-incubated with HeLa cells for 48 h. As shown in Figure 6A, a dose-dependent cytotoxicity effect could be observed for all drugs. The cell viability decreased to 52.3% (Cy), 16.1% (cisplatin), 59.3% (DHP) and 28% (DHP + Cy) at the highest concentration. However, the PVPt@Cy NPs did not show the highest cytotoxicity, the cell viability remained as high as 32.8% after 48 h of incubation, with the concentration of the Pt as high as 10.0 mg/L and Cy as high as 4.8 mg/L. The PVPt@Cy NPs possessed lower cytotoxicity, which might have been due to the slower rate of cellular uptake and the reduction in Pt (IV).

Furthermore, an MTT assay was also conducted to verify the phototoxicity of the PVPt@Cy NPs. The results are shown in Figure 6B. Certainly, compared with the corresponding dark groups, no distinct inhibitory effect was observed for the cisplatin or DHP groups treated with NIR irradiation. However, for the other light-treated groups (Cy, DHP+ Cy and PVPt@Cy NPs), the cell viabilities were obviously decreased. Specifically, nearly 84.9% of the HeLa cells were killed by the PVPt@Cy NPs with 808-nm laser irradiation. Moreover, the cell viability of the Cy and DHP groups was decreased to 35.6% and 57.9%, which were higher than that of the PVPt@Cy NPs group. These results indicate that the PVPt@Cy NPs not only reduced the systematic toxicity but also possessed synergistic anticancer potency and could serve an effective nanoplatform for cancer treatment.

### 4.7. In Vivo Biodistribution and Imaging

In addition to its use for PTT and PDT, Cy also could act as a medicament for diagnostics due to the inherent NIR fluorescence and thermal imaging ability. Initially, the in vivo NIR fluorescence imaging experiments were performed to evaluate the real-time biodistribution and tumor accumulation of the PVPt@Cy NPs. The H22 tumor-bearing mice were intravenously injected with free Cy and PVPt@Cy NPs. As depicted in Figure 7A, the time-dependent biodistribution of free Cy and PVPt@Cy NPs could be clearly observed. For the free Cy-treated group, at 5 h post-injection, the NIR fluorescence signal was the highest and then gradually decreased in the tumor site. However, in the mice treated with PVPt@Cy NPs, the fluorescence signal was weak at 5 h and then gradually enriched in the tumor site, and this signal was maintained for up to 24 h. The results revealed that the PVPt@Cy NPs had outstanding stability during blood circulation and perfect tumor accumulation properties.

We further examined the photothermal imaging performance of the PVPt@Cy NPs, and the H22 tumor-bearing mice treated with saline were used as a control. Based on the results of the in vitro photothermal and in vivo biodistribution experiments, we conducted laser irradiation (808 nm, 2 W/cm^2^, 4 min) for the mice at 24 h after the tail vein injections. The surface temperature of the mice with 808-nm laser irradiation was recorded in real time, and the images are shown in Figure 7B. Upon 808-nm light irradiation, for the PBS group, the highest tumor temperature increased to 42.8 °C within 4 min, while the temperature of the free Cy group rose from 37.2 °C to 46.3 °C and increased by 9.1 °C. Importantly, the temperature of the PVPt@Cy NPs group increased to 48.1 °C, which was enough to generate hyperthermia for effective photothermal therapy. Consequently, PVPt@Cy NPs have great potency as an ideal nanoplatform for imaging-guided chemo-phototherapy.

## 5. Conclusions

In summary, a multifunctional reduction-activated prodrug nanoplatform was prepared through a facial one-pot coupling reaction followed by self-assembly for combined chemotherapy, PTT and PDT. The self-assembled nanodrug PVPt@Cy NPs were pH-, reduction- and NIR light-responsive. The Pt (IV) acted as a hydrophobic segment and anticancer prodrug simultaneously, and the encapsulated Cy played multiple roles, including PTT agent, photosensitizer for PDT and imaging agent. Under 808-nm laser (1 W/cm^2^) irradiation, the temperature of the PVPt NPs (20 mg/L) increased by 11.6 °C within 300 s, and ROS was generated simultaneously. Meanwhile, in vitro studies have confirmed that PVPt@Cy NPs possess remarkable anticancer efficacy under synergistic chemotherapy, PTT and PDT. Moreover, in vivo fluorescence imaging and photothermal imaging demonstrated that the PVPt@Cy NPs could effectively enrich in the tumor site and generate enough heat. PVPt@Cy NPs are promising for imaging-guided combinational cancer therapy.

## Data Availability

Data is contained within the article or Appendix A.

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
