# Peer review of "NIR and Reduction Dual-Sensitive Polymeric Prodrug Nanoparticles for Bioimaging and Combined Chemo-Phototherapy"

_polymers, 2022, doi:10.3390/polym14020287_

Round 1
Reviewer 1 Report
On request of Polymers, I reviewed the paper entitled “ NIR/Reduction Dual-sensitive Polymeric Prodrug Nanoparti-2 cles for Bioimaging and Combined Chemo-phototherapy” by Shuying Li. The work deals with the synthesis of a multifunctional prodrug nanoparticles able to combine chemotherapy, photothermal therapy and tumor imaging. The work needs some adjustments before publication, as it should be more clearly exposed. Moreover the results are poorly discussed. In the introduction the Authors highlighted and cited many works focusing on the same topic, however they do not provide any comparison between their results and previous achievements, therefore the actual improvements that their work may have brought are not emphasized in a proper way. In the discussion it is necessary to include a comparison with other materials in order to highlight the differences and the advancements of the prepared material.
Specific comments:
- Please, check the affiliations and add the superscript number.
- Reference 1 does not deal with the success of chemotherapy in treating cancer but it is a research paper about a theranostic tool. I believe that a topic review on cancer treatment could be more appropriate.
- Again ref number 2 and 3 do not explain MDR or drug rapid metabolism and all the limitation of a conventional treatment. Please, select other more suitable papers.
- Scheme 1, please add in the caption the explanation of each abbreviation employed in the sketch.
- Line 80, VE stands for tochoferol
- Line 93, what was the cynine dye used? In materials the dye with the abbreviation of Cy was not reported.
- Line 113, use always L as measurement unit (mL).
- Line 132, The Authors must provide the weight of reagents and products, the volumes of all solvents, the reaction yields! All these details are mandatory in a Material and Methods section well-written.
- Line 215, more details regarding cell treatment are necessary: in which solvents were the samples dissolved? And what was the concentrations of loaded and empty NPs, dye and cisplatinum? Notably, the concentration of PVPt@Cy NPs employed which amount of cisplatinum and Cy provided? Can the Authors better explain the loading ratio of the dye and cisplatinum?
- Line 216 and Figure 6. DHP what does it mean? In material and methods this abbreviation is not mentioned.
- Figure 6, the Authors should also indicate in the text the molar concentrations of Cy and cisplatinum employed.
- Scheme 2. The synthesis is not clear, add more details such as solvents, time, purification method and yields.
Author Response
Response to Reviewer 1 Comments
Dear reviewer:
I really appreciate your thoughtful comments, which are very valuable for improving our paper. Special thanks to you for your well comments and suggestions. We have revised our manuscript one by one according to your comments and suggestions. For your comments, in the introduction, we added the some results of the cited works and compared them with previous achievements to highlight the practical improvements. In the results and discussions, comparisons with previous works have also been added to highlight differences and advances. Meanwhile, the following changes were made to specific comments:
Point 1: Please, check the affiliations and add the superscript number.

Response 1: Thank you very much for your comment. We rechecked the author affiliations, and confirmed correct. Moreover, the superscript number has been added and highlighted in red.
Point 2: Reference 1 does not deal with the success of chemotherapy in treating cancer but it is a research paper about a theranostic tool. I believe that a topic review on cancer treatment could be more appropriate.
Response 2: According to the comment, we replaced reference 1 with a more appropriate reference and highlighted it in red.
Point 3: Again ref number 2 and 3 do not explain MDR or drug rapid metabolism and all the limitation of a conventional treatment. Please, select other more suitable papers.
Response 3: We have chosen new references according to your suggestion. The revisions were highlighted in red.
Point 4: Scheme 1, please add in the caption the explanation of each abbreviation employed in the sketch.
Response 4: Thanks for your advice.
In scheme 1, we have improved presentation of various abbreviations and added the explanation of each abbreviation employed in the sketch in the caption as follows:
Scheme 1. Schematic illustration of the self-assembly of PVPt@Cy NPs, and the intracellular action after endocytosis for the combined chemo-phototherapy. Description of abbreviations in the scheme, PMA: poly(isobutylene-alt-maleic anhydride); Pt(II): actived platinum(II) anticancer drug; Cy: HOCyOH, a cyanine dye; mPEG2k-OH: methoxypolyethylene glycol.
Point 5: Line 80, VE stands for tochoferol
Response 5: In line 89, α-parcopherol was replaced with α-tochoferol according to comment and we added the explanation of α-tochoferol in materials. At the same time, VE was replaced with α-tochoferol, we highlighted them in red.
Point 6: Line 93, what was the cynine dye used? In materials the dye with the abbreviation of Cy was not reported.
Response 6: According to the previous literature[36], we synthesized a dye 1-(2-hydroxyethyl)-2-((E)-2-((E)-3-((E)-2-(1-(2-hydroxyethyl)-3,3-dimethylindolin-2-ylidene)ethylidene)-2-chlorocyclohex-1-en-1-ly)vinyl)-3,3-dimethyl-3H-indol-1-ium bromide based on cyanine dye modification, and expressed as Cy (line 84 and 119). Please refer to the support information for specific synthesis routes. Meanwhile, according to the comments, we added explanation of Cy in materials, and the revised text was marked in red.
Point 7: Line 113, use always L as measurement unit (mL).
Response 7: Thank you very much for your comment. We have replaced all mL with L as the measurement units in the text according to comment and highlighted them in red.
Point 8: Line 132, The Authors must provide the weight of reagents and products, the volumes of all solvents, the reaction yields! All these details are mandatory in a Material and Methods section well-written.
Response 8: Thank you very much for your comment.
We added the details of raw materials and product used in the experiment, such as the volume and weight of the reagent, the weight and yield of the synthesized product (line 148), details of the changes were highlighted in red.
Point 9: Line 215, more details regarding cell treatment are necessary: in which solvents were the samples dissolved? And what was the concentrations of loaded and empty NPs, dye and cisplatinum? Notably, the concentration of PVPt@Cy NPs employed which amount of cisplatinum and Cy provided? Can the Authors better explain the loading ratio of the dye and cisplatinum?
Response 9: First, all the drugs were dissolved or suspended in DMEM medium, and the concentrations were located in the Figure 6. According to your suggestion, the concentration of drugs was added to the methods section as follows:
After incubation overnight, the medium was replaced with fresh DMEM medium, which contained various drugs at Pt concentrations (0.31, 0.63, 1.25, 2.5, 5, 10) or Cy concentrations (0.15, 0.3, 0.6, 1.2, 2.4, 4.8) (mg/L), separately.
Point 10: Line 216 and Figure 6. DHP what does it mean? In material and methods this abbreviation is not mentioned.
Response 10: Thank you for your suggestion.
DHP was synthesized according to previous literature [37, 38] (Figure S1.), and we added DHP in materials. Meanwhile, the revised text was marked in red.
Point 11: Figure 6, the Authors should also indicate in the text the molar concentrations of Cy and cisplatinum employed.
Response 11: Thank you very much for your comment.
In Figure 6, we marked different concentrations of Cy and Pt. Meanwhile, we added the concentration of Cy and Pt employed in the text (Line 221) for readers to better understand and marked them in red according to the comment.
Point 12: Scheme 2. The synthesis is not clear, add more details such as solvents, time, purification method and yields.
Response 12: We have revised the synthesis route in Scheme 2. Meanwhile, we have added more details in text (Line 265). The revised scheme 2 is as follows:

Reviewer 2 Report
Paper titled (NIR/Reduction Dual-sensitive Polymeric Prodrug Nanoparticles for Bioimaging and Combined Chemo-phototherapy) by Li et al. described the synthesis and characterization of a novel prodrug nanoparticles and testing it is activity in cell lines and an in vivo model.
1- Methods: I suggest dividing it into 2 main parts (1&2) and then subtitles. Part 1. Synthesis and formulation of the ...............nanoparticles & Part2 : Biological activity. This will be more organized and easier for the reader.
2- In general, the methods lack references. please add references where ever adequate.
3- Authors must consider statistical analysis for the results.
4- Please add a section in methods: describe that what test was applied for checking normality of data distribution and what statistical tests were applied for comparison of the groups & pair-wise comparison. What software + version & what was the p value.
5- In all figure legends, make sure that the abbreviations used in the figure itself are described.
6-Fig 4 B, C & E: needs statistical analysis at least at the end point.
7-Fig 5A: needs analysis for the last point in the curve
8- Fig 6A&B: needs stat analysis & expand each panel in full page width to be easy for the reader
9- Animal experiment: what was the name of the regional animal committee & what is the number & date of approval
9-
Author Response
Response to Reviewer 2 Comments
Point 1: Methods: I suggest dividing it into 2 main parts (1&2) and then subtitles. Part 1. Synthesis and formulation of the ...............nanoparticles & Part2 : Biological activity. This will be more organized and easier for the reader.
Response 1: Thank you for your recommendation. In order to make the article more organized and easier for readers to understand, “Methods” has been divided into two main parts and then subtitle according to the comments. Part 1: Synthesis and formulation of the PVPt@Cy NPs, which includes the preparation and basic characterization of PVPt@Cy NPs. Part 2: Biological activity, which contains the representation of Biological level, and marked the revision in red.
Point 2: In general, the methods lack references. please add references where ever adequate.
Response 2: According to the comments, we supplemented relevant literature in methods, and marked red in Reference.
Point 3: Authors must consider statistical analysis for the results.
Response 3: Thank you for your suggestion. We have added the statistical analysis of the results (Figure 4-6) in the Results and Discussion section, and marked it in red.
Point 4: Please add a section in methods: describe that what test was applied for checking normality of data distribution and what statistical tests were applied for comparison of the groups & pair-wise comparison. What software + version & what was the p value.
Response 4: According to your suggestion, we have revised the statistical analysis in the method section. The revision was highlighted in red.
Point 5: In all figure legends, make sure that the abbreviations used in the figure itself are described.
Response 5: We rechecked the manuscript, added the full name of the missing abbreviation, and marked it in red.
Point 6: Fig 4 B, C & E: needs statistical analysis at least at the end point.
Response 6: Thank you very much for your appropriate suggestions. In Figure 4B, C&E, We selected the average value of the three groups of data to get the results. According to the comments, the original three sets of data are selected for statistical analysis as follows, and the modified figures have been marked in red in the manuscript.
Point 7: Fig 5A: needs analysis for the last point in the curve
Response 7: In Figure 5A, we conducted statistical analysis according to the comments, and the revised figure has been marked in red in the manuscript.
Point 8: Fig 6A&B: needs stat analysis & expand each panel in full page width to be easy for the reader
Response 8: In Figure 6A&B, we added the stat analysis and highlighted it in red in the manuscript.
Point 9: Animal experiment: what was the name of the regional animal committee & what is the number & date of approval
Response 9: The name of the Regional Animal Committee is the First Affiliated Hospital, Sun Yat-sen University and the approval number is 202010115. The date is October 11, 2020. In addition, the animal experiments was conducted by Shuying Li in Southern Medical University with the assistance of Wang Yupeng.

Round 2
Reviewer 1 Report
The authors satisfied all requests no further corrections are needed except one.When I asked to change the unit of measurement (L), I did not intend to transform all mL into L but ml into mL.
Consequently the authors must correct the text in this sense.
Author Response
Point 1: The authors satisfied all requests no further corrections are needed except one.
When I asked to change the unit of measurement (L), I did not intend to transform all mL into L but ml into mL.
Consequently the authors must correct the text in this sense.

Response 1: Thank you again for your comment and suggestion. We have revised the manuscript again, and replaced the ml in the manuscript with mL. The suitable units were further adopted. All the revisions were highlighted in red.

Reviewer 2 Report
Thanks for authors for revising the manuscript, I still have 2 comments:
1- Fig 6 legend: data nor date
2- In all figures, write what was the stat test applied & what * denotes difference from which group?
3- Fig 6 A: conc. 5, there is a *** sympol, please revise it, it is unexpected to be like that between the assigned 2 groups
4- AUthors must clarify in each figure what is the sympols refer to!! I mean difference from which group exactly.
Author Response
Response to Reviewer 2 Comments
Point 1: Fig 6 legend: data nor date
Response 1: Thank you again for your comment, we corrected the spelling of “data” and marked it in the manuscript.
Point 2: In all figures, write what was the stat test applied & what * denotes difference from which group?
Response 2: According to the comments, we have added the student’s t-test applied in all figures, meanwhile, we added the differences of which groups are represented by “ ns ,*, **, *** and ****”.
Point 3: Fig 6 A: conc. 5, there is a *** sympol, please revise it, it is unexpected to be like that between the assigned 2 groups
Response 3: We have carefully checked the statistical differences added in Figure 6, corrected and highlighted them in red.
Point 4: AUthors must clarify in each figure what is the sympols refer to!! I mean difference from which group exactly.
Response 4: Thank you again for your advantageous comments. We added the explanation of the statistical difference symbol in the figure and highlighted it in red.

Round 3
Reviewer 2 Report
I recommend acceptance of the current form